# Microstructure Evolution in Plastic Deformed Bismuth Telluride for the Enhancement of Thermoelectric Properties

**DOI:** 10.3390/ma15124204

**Published:** 2022-06-14

**Authors:** Haishan Shen, In-Yea Kim, Jea-Hong Lim, Hong-Baek Cho, Yong-Ho Choa

**Affiliations:** 1Department of Materials Science and Chemical Engineering, Hanyang University, 55 Hanyangdaehak-ro, Sangnok-gu, Ansan 15588, Korea; seadheart@hanyang.a.kr (H.S.); hongbaek@hanyang.ac.kr (H.-B.C.); 2Department of Materials Science and Engineering, Gachon University, 1342 Seongnamdaero, Sujeong-gu, Seongnam-si 13120, Korea; kiy7484@gachon.ac.kr (I.-Y.K.); limjh@gachon.ac.kr (J.-H.L.)

**Keywords:** n-type Bi_2_Te_3_, powder processing, cold pressing, canning package, recycled waste scraps

## Abstract

Thermoelectric generators are solid-state energy-converting devices that are promising alternative energy sources. However, during the fabrication of these devices, many waste scraps that are not eco-friendly and with high material cost are produced. In this work, a simple powder processing technology is applied to prepare n-type Bi_2_Te_3_ pellets by cold pressing (high pressure at room temperature) and annealing the treatment with a canning package to recycle waste scraps. High-pressure cold pressing causes the plastic deformation of densely packed pellets. Then, the thermoelectric properties of pellets are improved through high-temperature annealing (500 ${}^{\circ }$C) without phase separation. This enhancement occurs because tellurium cannot escape from the canning package. In addition, high-temperature annealing induces rapid grain growth and rearrangement, resulting in a porous structure. Electrical conductivity is increased by abnormal grain growth, whereas thermal conductivity is decreased by the porous structure with phonon scattering. Owing to the low thermal conductivity and satisfactory electrical conductivity, the highest ZT value (i.e., 1.0) is obtained by the samples annealed at 500 ${}^{\circ }$C. Hence, the proposed method is suitable for a cost-effective and environmentally friendly way.

## 1. Introduction

Thermoelectric generators (TEGs) can directly convert thermal energy to electricity. They are excellent materials for renewable energy applications to reduce the environmental impact of CO_2_ emissions and achieve net-zero emissions by 2050 under the Paris Agreement [1,2]. These materials recover waste heat from sources, such as manufacturing plants, combustion engines, and even the human body, and convert it to electrical energy. In addition, they are noiseless, have a long service life, and do not require large-scale systems. Accordingly, TEGs have a wide range of possible applications, including Internet-of-Things sensors, wearable devices, internal combustion engine vehicles, and manufacturing plants [3,4,5]. The performance of TEGs is typically evaluated using the thermoelectric figure of merit (ZT), defined as
(1)$$ZT=\frac{\alpha^{2}\sigma }{\kappa }T,$$
where α is the Seebeck coefficient (µV/K); σ is the electrical conductivity (S/m); κ is the thermal conductivity (W/(m·K)); and T is the absolute temperature (K). A high ZT value indicates that thermoelectric materials have a high electric conductivity and Seebeck coefficient; however, their thermal conductivity is low. Many researchers endeavor to improve the ZT value by reducing the thermal conductivity of materials (e.g., phonon-glass electron crystals [6,7] and nanostructured material [8,9,10,11]), conducting electronic band engineering [12,13], examining the energy-filtering effect and defect [14,15,16], and implementing grain boundary engineering [17,18,19]. Bismuth–tellurium-based materials are excellent TEGs for near-room temperature (RT) applications and are widely used commercially. Typically, the fabrication of TEG devices involves a series of processes, such as the synthesis of bulk ingots via zone melting, dicing, plating cleaning, and soldering [20,21,22,23]. During device fabrication, expensive material resources are wasted because many scraps are unavoidably produced [24,25,26]. The recycling of waste scraps by grinding them into powders and then synthesizing them into high-performance samples is an excellent industrial achievement with reduced costs, energy consumption, and environmental friendliness. Conventionally, the powder sintering process involves hot pressing (HP) [27,28,29] and a hot-pressing texture (HPT) [27,30] to achieve fully dense ceramics. However, the cost and contamination limit the general applicability of the process to a few ceramic systems [31]. Recently, to compact powders, spark plasma sintering (SPS) has been applied to many Bi–Te-based materials [19,28,32,33] because it enables rapid heating and cooling. However, the foregoing techniques consume considerable energy and are sometimes applied in a vacuum system. In contrast, cold sintering implemented at RT and high pressure (hundreds of megapascals to gigapascals) produces high-density pellets without the necessity of energy-consuming technologies, such as HP, HPT, and SPS [34,35,36]. In this work, we recycled commercial n-type Bi_2_Te_3_ scraps and synthesized the compounds by cold pressing (high pressure at room temperature), followed by annealing treatment using a coin cell canning package. Moreover, optimizing the annealing treatment using a canning package improved the thermoelectric performance. Because the electrical conductivity was excellent and the thermal conductivity was low, the highest ZT value (i.e., 1.0) was obtained at a high annealing temperature (500 ${}^{\circ }$C) without phase deformation. The foregoing technique can be applied to mass production with no energy consumption.

## 2. Experiments

### 2.1. Experimental Section

For this study, n-type thermoelectric waste scraps were obtained from a commercial Bi_2_Te_3_ ingot (Kryotherm Co., Ltd., Saint Petersburg, Russia). The scraps were ground into fine powder using a mortar. To exclude the effect of particle size on the thermoelectric properties of the material during the sintering process, the powdered scraps were sieved such that the particle size was between 45 and 53 µm. At RT, the sieved powder was loaded into a hydraulic press die (Ø10 mm) with high-pressure compression (1.5 GPa) for 5 min to produce a highly dense compacted pellet with approximately 1 mm of thickness. To avoid phase separation, a coin cell canning package between two graphite foils was utilized due to the thermal and chemical stabilities of graphite foil (Figure 1). The pellets packed by canning were annealed at different temperatures (300, 400, and 500 ${}^{\circ }$C) for 1 h in a tube furnace with Ar gas flow to enhance the thermoelectric properties.

### 2.2. Characterization

The crystal structure and morphology of the pellets were analyzed by X-ray diffraction (XRD, Cu Kα, Rigaku Co., Tokyo, Japan) at 40 kV and 100 mA. The morphology of the pellets was observed by field-emission scanning electron microscopy (FE-SEM, Hitachi Ltd., Tokyo, Japan). High-resolution images and microstructure morphologies were obtained using transmission electron microscopy (TEM, JEM-2100F, JEOL, Tokyo, Japan) at 200 kV. To further analyze the microstructure, electron backscattered diffraction (EBSD, Velocity Super, Ametek, Berwyn, PA, USA) was implemented using a field emission gun scanning electron microscope (SU5000, Hitachi, Tokyo, Japan). All EBSD data were analyzed using the TSL-OIM software. Data points with a confidence index of less than 0.1 were removed from the EBSD data. Thermoelectric properties (electrical and Seebeck coefficient) of pellets were measured in an in-plane direction, which possesses a high performance [37]. Electrical conductivity (σ) was measured by the four-point probe method (CMT-SR 1000N, Advanced Instrument Technology, Suwon, Korea) and hall measurement (HMS-5000, Ecopia, Chandler Hall, AZ, USA) at room temperature. Before measuring the electrical conductivity, the pellet thickness was determined using a micrometer caliper (MDC-25MJ, Mitutoyo, Kawasaki, Japan). The Seebeck coefficient (α) was measured at RT using customized measurement techniques. The total thermal conductivity (κ) was calculated as κ = DC${}_{p}$ρ, where D, C${}_{p}$, and ρ are the thermal diffusivity coefficient, specific heat capacity, and density, respectively. The thermal diffusivity coefficient was measured using a laser flash apparatus (Netzch LFA 467, NETZCH, Selb, Germany), and the specific heat capacity was measured using a differential scanning calorimeter (DSC-60 plus, Shimadzu, Kyoto, Japan). The pellet densities were determined using an immersion technique (Archimedes principle) at RT with ethanol as a medium.

## 3. Results and Discussions

The XRD patterns of the ingot at RT and annealed Bi_2_Te_3_ pellets (with and without coin cell canning packages) are shown in Figure 2. For the pellets without canning packages, phase separation was observed as the annealing temperature increased. In particular, for the sample without the coin cell package, Bi${}_{4-x}$Te${}_{3+x}$ peaks appeared at 500 ${}^{\circ }$C (annealing temperature). This explains the dissociation and sublimation of tellurium during annealing at high temperatures (Figure 2a) [38]. The peak split at (1 0 10) of the XRD pattern was detected at annealing temperatures of 400 and 500 ${}^{\circ }$C, indicating that phase separation occurred at 400 ${}^{\circ }$C and above (Figure 2c). Correspondingly, no phase separation or other peaks were observed in the pellet of the sample with the coin cell canning package, although the pellets were annealed at a high temperature value (500 ${}^{\circ }$C) (Figure 2b,d). This indicates that the closely packed graphite foil surrounding the cold-pressed pellet blocked the escape of tellurium from the samples. Interestingly, the RT sample exhibits slight peak shifts from 37.92${}^{\circ }$ to 37.96${}^{\circ }$, 38.0${}^{\circ }$, and 38.02${}^{\circ }$ at 300, 400, and 500 ${}^{\circ }$C, respectively. Moreover, the peak at 500 ${}^{\circ }$C virtually matches the ingot peak (i.e., 38.04${}^{\circ }$). This may be the result of the residual compressive strain and stress in the test sample due to the applied high pressure and the release of residual energy via the annealing treatment. For the cold-pressed sample prepared at RT (i.e., no annealing) and pellets annealed at 300 ${}^{\circ }$C, the main peaks are at (0 0 6); their intensity was considerably higher than that of the peak at (0 1 5) (the main peak of Bi_2_Te_3_ and ingot based on standard data (JCPDS no. 85-439)). The relative peak intensity ratios (I_006_/I_015_) are 1.237, 1.105, 0.839, and 0.836 at RT, 300, 400, and 500 ${}^{\circ }$C (with the canning package), respectively. These intensity ratios are considerably higher than that of ingot (i.e., 0.775). These results suggest that the cold-pressed pellet subjected to a 1.5-GPa pressure value preferred the (0 0 l) orientation perpendicular to the press direction by high-pressure stress. After the annealing process, the (0 0 l) texture orientation decreased, and the intensity of the main peak of (0 1 5) increased with the annealing temperature. Such a texture reorientation is attributed to the recrystallization process, releasing residual energy from cold pressing [39]. Jun et al. observed the same recrystallization process at the annealing treatment when they applied strain energy to stoichiometric ingots (Bi${}_{0.45}$Sb${}_{1.55}$Te${}_{3}$) by cold pressing (pressure at gigapascal scale) and then annealing at 300 ${}^{\circ }$C in a vacuum [39].

The top view and FE-SEM images of the fractured surfaces of specimens prepared at RT, 300, 400, and 500 ${}^{\circ }$C are shown in Figure 3. The top view shows numerous pores and microcracks on the surface of the RT pellet, indicating that interparticle bonds by van der Waals forces are produced via high pressure without any other heat energy. When heat energy was introduced into the cold-pressed pellet, the microcracks disappeared, and the number of large pores were decreased leading to the smoother surface morphology (Figure 3a–d). For the fractured surface, lamellar grains were densely stacked parallel to the pressure direction. As heat energy was applied (i.e., by annealing) to the cold-pressed pellets, grain growth and grain realignment occurred in the annealed samples; the grain growth was considerable with increasing annealing temperature. In addition, many pores among the grains are observed in the pellet annealed at 500 ${}^{\circ }$C; this is consistent with the top view of surface images. The foregoing leads to the conclusion that the densely packed pellets (cold-pressed at high pressure) display rapid grain growth involving the disappearance of pores and microcracks during the annealing process. However, the high-temperature annealing contributes to rapid grain growth and grain realignment, generating many pores and low-density pellets, as shown by the samples annealed at 300 and 400 ${}^{\circ }$C.

The density of each sample was also measured during annealing (with and without the canning process). All pressed pellets have high relative densities exceeding 90% with respect to the ingot density of 7.721 g/cm${}^{3}$ (Appendix A). For the pellets prepared at RT, the relative density reached 97.9%, indicating that they were densely packed at a high pressure. Further, the relative density of pellets slightly increased at 300 ${}^{\circ }$C and then decreased at 400 and 500 ${}^{\circ }$C. This trend was observed regardless of the canning package and agreeing well with the FE-SEM images, in which some pores and microcracks shrunk during the abnormal grain growth at low-temperature annealing. The rapid grain growth and grain realignment during high-temperature annealing led to the generation of pores and low-density pellets.

The effect of annealing on the electrical conductivity at various annealing temperatures is shown in Figure 4a. As the annealing temperature increased, the electrical conductivity gradually increased from 3.45 ± 0.04 × 10${}^{4}$ (at RT) to 12.22 ± 1 × 10${}^{4}$ S/m (at 500 ${}^{\circ }$C) due to the grain growth induced by annealing. In Appendix A, the hall measurement about electrical conductivity also displayed the same trend with 4-probe measurement. The electron mobility at room temperature increased from 56 to 101.22 cm${}^{2}$/V·s, implying the grain growth for the pellets. The carrier concentration decreased at 300 ${}^{\circ }$C and increased again for high annealing treatment pellets. The Seebeck coefficient of the annealing-treated pellet was slightly smaller than that of the cold-pressed pellet (Figure 4b). The thermal conductivity first increased from 0.61 (at RT) to 1.63 W/m·K (at 400 ${}^{\circ }$C) and then decreased to 1.17 W/m·K at 500 ${}^{\circ }$C (Figure 4c). This may be due to the grain growth and presence of large grains, which induce less electron scattering, leading to high electrical and thermal conductivities. However, the porous structure in pellets may cause considerable phonon scattering, causing the thermal conductivity at 500 ${}^{\circ }$C to be lower than that at 400 ${}^{\circ }$C. Owing to the significantly lower thermal conductivity (0.6 W/m·K), at RT, the ZT value is slightly higher than those of the pellets annealed at 300 and 400 ${}^{\circ }$C. Then, the pellet annealed at 500 ${}^{\circ }$C, the ZT value substantially increased to approximately 1.0, leading to high electrical conductivity and low thermal conductivity. These results indicate that dense cold-pressed pellets compacted by high pressure have numerous microcracks that facilitate low electrical and thermal conductivities due to electron scattering. However, with grain growth and grain realignment during annealing, the microcracks disappeared, thus increasing the electrical and thermal conductivities. In addition, rapid grain growth and grain realignment created porous pellets that reduced the thermal conductivity by phonon scattering.

The crystal orientation mapping images from the EBSD data, with an image quality (IQ) map, a color-coded inverse pole figure (IPF) map, and grain size distribution, clearly verify the present state. The overall IQ map shown in Figure 5a distinctly exhibits the grain morphology of each sample. For the samples at RT and annealed at 300 ${}^{\circ }$C, elongated and non-equiaxed grains were observed. The dark gray shades indicate the grains that are considerably deformed by high pressure and may be highly compacted by small grains. Because of the high pressure in the z-axis direction, the elongated grains preferred the [0 0 1] orientation, as shown in Figure 5b: the red, green, and blue colors represent the [0 0 1], [−1 −2 0], and [1 −1 0] directions, respectively. Moreover, the maximum intensity of the (0 0 l) plane was 13.11 for the samples prepared at RT; the intensity decreased to 8.29 for the samples annealed at 300 ${}^{\circ }$C (Appendix A). However, the grains of the samples annealed at 400 and 500 ${}^{\circ }$C were recrystallized by grain growth and changed to random orientation (Figure 5b). The maximum intensity changed to 4.012 at 400 ${}^{\circ }$C and then increased to 7.227 in the (0 1 5) plane at 500 ${}^{\circ }$C (Appendix A). The grains were recrystallized during the annealing process and changed to a random orientation. In particular, at high annealing temperatures, the grains rapidly grow and rearrange in the (0 1 5) plane, which is the main plane of Bi_2_Te_3_. Because of the rapid grain growth, a porous structure is observed in the IQ map of the samples annealed at 500 ${}^{\circ }$C without applied pressure (Figure 5a) [40,41].

The grain size distribution in the samples that have been cold-pressed and annealed at different temperatures after cold pressing is shown in Figure 5c. Most of the grain sizes are less than 1 µm. In the sample cold-pressed at RT, the grain size exceeds 30 µm. These large grains may be elongated due to the distinct local plastic deformation caused by high pressure. The TEM analysis results shown in Appendix A also indicate the existence of a lamellar structure with elongated subgrains perpendicular to the pressing direction. The highly distributed low-angle grain boundaries in the sample prepared at RT also exhibit local plastic deformation, as shown in Appendix A [42,43]. These results are based on the unit cell of Bi_2_Te_3_ that is a trigonal crystal structure consisting of quintuple atomic stacked layers of Bi and Te atoms in the sequence Te${}^{\left(1\right)}$–Bi–Te${}^{\left(2\right)}$–Bi–Te${}^{\left(1\right)}$ [44,45,46]. The five stacked atomic layers interact by van der Waals forces because of the facile plastic deformation by basal slip [47,48,49]. In addition, thermo-mechanically treated polycrystalline bulk thermoelectric materials introduce dislocations through plastic deformation. These dislocations improve the densification and control the crystallographic texture of the material, as demonstrated by the case of samples prepared at RT [47]. In contrast, in the less-compacted area packed with small random grains, large pores and microcracks were detected at the interfacial grains (Appendix A). Overall, in the absence of heat energy, the particles were physically compressed by high pressure through plastic deformation; further, the formation of pores and microcracks resulted in a densely compacted sample.

After the introduction of heat energy, the plastically deformed grains began to coarsen. Consequently, the degree of distribution of low-angle grain boundaries slightly decreased at 300 ${}^{\circ }$C. Then, at 400 and 500 ${}^{\circ }$C, the samples gained higher angle grain boundaries (such as 60°) (Appendix A). At high annealing temperatures (400 and 500 ${}^{\circ }$C), the main grain size distribution trend is bimodal, indicating abnormal grain growth (Figure 5c) [50,51]. The kinetics of abnormal growth is based on the recrystallization and dynamic recovery from the plastic deformation that occurs at elevated temperatures [52,53]. In some instances, grain boundaries were also observed at the interfaces between two large grains; this explains the abnormal grain growth upon isothermal annealing, as shown in Appendix A. However, at high temperatures, rapid grain growth and nanosized pores were detected in the grains (Appendix A). Note that this grain size increase was more rapid than the pore elimination during the abnormal grain growth at high annealing temperatures. This growth leads to the coalescence of large grains and production of pores along the large grains; this is consistent with the SEM and EBSD images.

In summary, our results strongly indicate that the pristine powders were highly compacted by plastic deformation under high pressure; then, the particles were coarsened through annealing. At high annealing temperatures, rapid grain growth results in the generation of porous structures (Figure 6a). As shown in Figure 6b, the texture orientation from the (0 0 l) plane in the samples highly compacted at RT is changed into that of random structures, implying grain recrystallization. Finally, the recrystallized subgrains underwent abnormal grain growth at high annealing temperatures for crystallization oriented at (0 1 5). Most Bi_2_Te_3_ thermoelectric materials (mainly oriented at (0 1 5)) have high thermoelectric properties because they are anisotropic [54,55,56]. Hence, the electrical conductivity of samples annealed at 500 ${}^{\circ }$C was considerably higher than those of the other samples although their porous structure affected electron scattering. As a result, the best thermoelectric properties were obtained from the samples treated at 500 ${}^{\circ }$C due to their low thermal conductivity and satisfactory electrical conductivity. This results from the grain rearrangement and grain growth through the annealing treatment with canning package after high-pressure cold pressing.

The ingot parameters and various powder processing methods on n-type Bi_2_Te_3_ pellets reported in previous studies are compared in Table 1. Our work demonstrates that low thermal conductivity is achieved because of the porous structure resulting from grain rearrangement. Moreover, high electrical conductivity via abnormal grain growth is realized. The simple and rapid powder processing technique consumes less energy than other methods. Therefore, it is suitable for mass production and cost-effective fabrication.

## 4. Conclusions

This study demonstrates the recycling of n-type Bi_2_Te_3_ waste scraps via a simple powder processing method involving cold pressing at high pressure and annealing treatment with canning package. The samples pressed by high pressure at RT have a (0 0 l) texture orientation with elongated grains and high packing density caused by the local plastic deformation. Subsequently, different annealing temperatures were applied to enhance the thermoelectric properties through the canning process. The sample that underwent the canning package (compared with that without this process) exhibited no phase transformation despite the high annealing temperature. The highest ZT value, i.e., 1.0, was attained by the samples annealed at 500 ${}^{\circ }$C. The rapid grain growth and rearrangement induced a porous structure, leading to low thermal conductivity through phonon scattering without a decrease in electrical conductivity. The technique proposed in this work is deemed to be a new and advantageous approach to achieve a cost-effective and environmentally friendly production.

## Figures and Tables

**Figure 1 materials-15-04204-f001:**
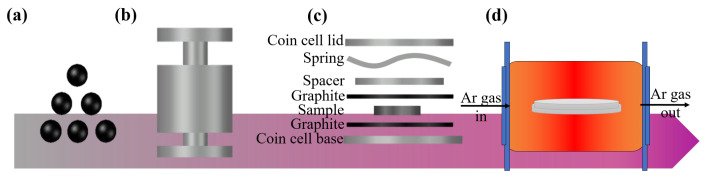
Schematic of experimental process: (**a**) powder grinding and sieving (45–53 µm); (**b**) cold pressing; (**c**) coin cell canning package; and (**d**) annealing process.

**Figure 2 materials-15-04204-f002:**
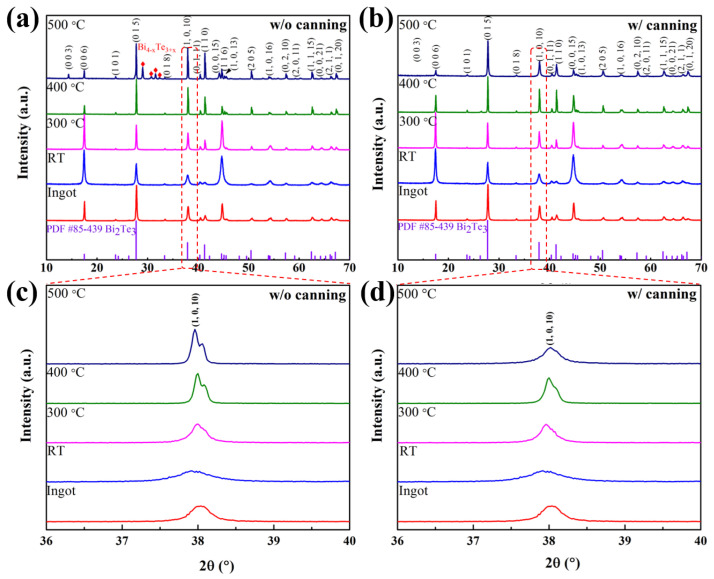
XRD patterns of ingot, Bi_2_Te_3_ pellet prepared at RT, and pellets annealed at various temperatures (300, 400, and 500 ${}^{\circ }$C) (**a**) without and (**b**) with coin cell canning package; magnified view of peaks at (1 0 10) (**c**) without and (**d**) with canning package accompanied by peak shift.

**Figure 3 materials-15-04204-f003:**
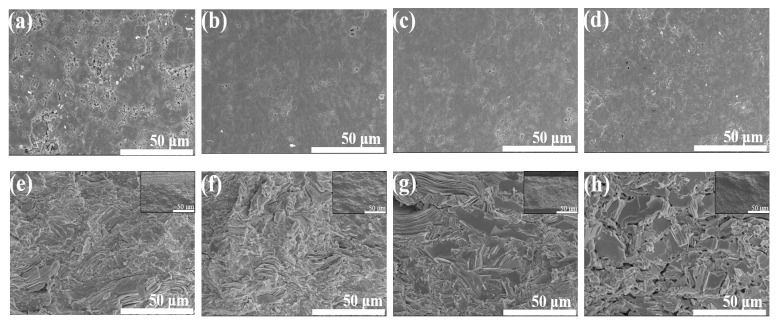
(**a**–**d**) Top view images; (**e**–**h**) fractured surfaces and full FE-SEM images of samples at RT, 300, 400, and 500 ${}^{\circ }$C.

**Figure 4 materials-15-04204-f004:**
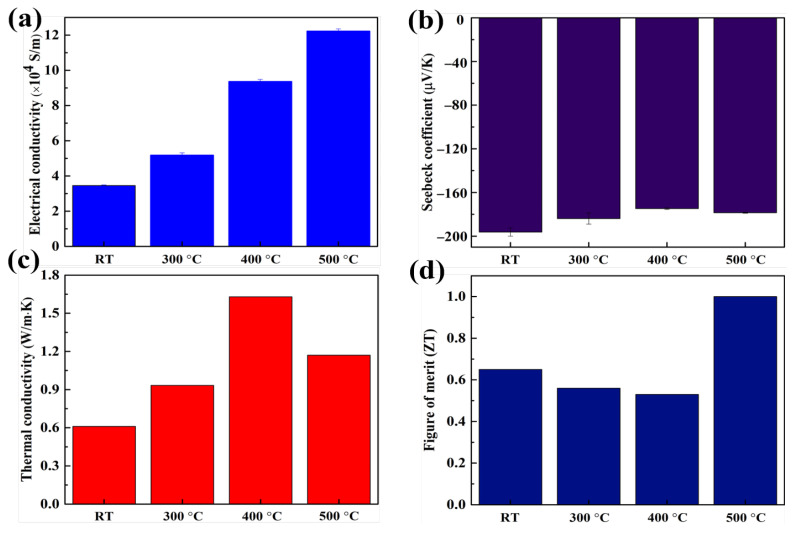
Measurement of thermoelectric properties at RT: (**a**) Electrical conductivity; (**b**) Seebeck coefficient; (**c**) thermal conductivity; and (**d**) ZT at RT, 300, 400, and 500 ${}^{\circ }$C.

**Figure 5 materials-15-04204-f005:**
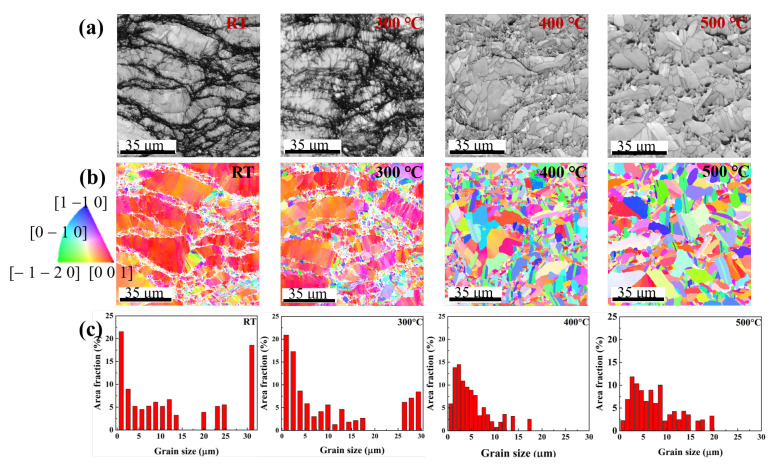
EBSD data with (**a**) IQ map, (**b**) color-coded IPF map, and (**c**) grain size distribution at RT, 300, 400, and 500 ${}^{\circ }$C.

**Figure 6 materials-15-04204-f006:**
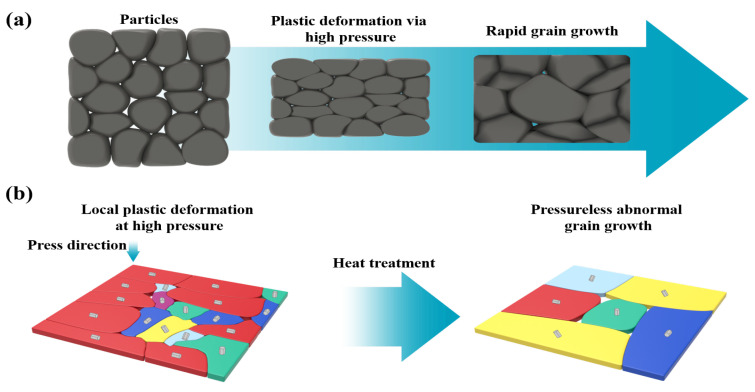
Schematic of (**a**) particle deformation and (**b**) texture change and grain growth from powder compaction with cold sintering at high pressure to rapid grain growth via high-temperature annealing.

**Table 1 materials-15-04204-t001:** Comparison of thermoelectric parameters of n-type Bi_2_Te_3_ pellet with various powder processing methods. All the thermoelectric properties and ZT were measured and obtained at room temperature.

Sample with Process	σ (×10${}^{5}$ S/m)	α (µV/K)	κ (W/m·K)	ZT	Reference
Bulk material (zone melting)	1.9	−180	1.8	1.03	From ingot
Microwave-activated hot-press sintering	0.56	−160	1.1	0.39	[29]
SPS and hot-forging	1.67	−140	1.15	0.85	[33]
Plasma-activated sintering	1.65	−125	1.38	0.56	[28]
High-pressure sintering	1.00	−147	0.8	0.81	[49]
Cold pressing and canning	1.22	−179	1.17	1.0	This work

## Data Availability

Not applicable.

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
