# Peer review of "Microstructure Evolution in Plastic Deformed Bismuth Telluride for the Enhancement of Thermoelectric Properties"

_materials, 2022, doi:10.3390/ma15124204_

Round 1
Reviewer 1 Report
In this manuscript, authors have investigated the evolution of microstructure in plastic deformed bismuth telluride for the enhancement of thermoelectric properties. Here, a simple powder processing technology is applied to prepare n-type Bi2Te3 pellets by cold pressing and annealing treatment with canning package to recycle waste scraps. The approach described in this paper is thought to be novel and advantageous. The crystal orientation mapping images from the EBSD data is quite impressive which supports their claim. However, there are a couple major concerns:
1. “However, the foregoing techniques consume considerable energy and are unable to apply high pressure.” This statement is not correct because in SPS also it is possible to apply high pressure. Author should modify this line.
2. 2. Modify figure 2 a and 2b. The overlap between temperature and indexing can be removed.
3. 3. What is the carrier concentration and mobility of the sample with temperature?
4. 4. Major concern: I guess author over/under estimated the ZT value because electrical conductivity and thermal conductivity were measured in two different direction. Author should correct this in the revised manuscript. What is the reproducibility of the data? I am not sure about the ZT value of this work in the present form because of the anisotropic structure of Bi2Te3.
5. 5. Why author did not increase the temperature more than 500-degree C for their study?
Author Response
Dear Editor,
I am pleased to submit a revised manuscript for publication in Materials. The title of the manuscript and the names of the authors are as follow:
Microstructure evolution in plastic deformed bismuth telluride for the enhancement of thermoelectric properties
Haishan Shen , In Yea Kim , Jea Hong Lim, Hong-Baek Cho , Yong-Ho Choa
We found that comments of the reviewers were very helpful in reshaping the paper. We revised the manuscript accordingly. Below is the list of our responses to the comments.
Responses to Reviewer’s comments
Point 1: “However, the foregoing techniques consume considerable energy and are unable to apply high pressure.” This statement is not correct because in SPS also it is possible to apply high pressure. Author should modify this line.
Response 1: In order to comply with the referee’s comment, we rewrote and edited the other SPS disadvantage in line 37 as following.
From: However, the foregoing techniques consume considerable energy and are unable to apply high pressure.
To: However, the foregoing techniques consume considerable energy and sometimes apply in vacuum system.
Point 2: Modify figure 2a and 2b. The overlap between temperature and indexing can be removed.
Response 2: We modified the figure 2.
Point 3: What is the carrier concentration and mobility of the sample with temperature?
Response 3: We add the hall measurement information in Fig. S2 and line 143~147.
Added text (line 143~147): In Fig. S2, the hall measurement about electrical conductivity also displayed the same trend with 4-probe measurement. The electron mobility at room temperature increase from 56 to 101.22 cm2/V·s, implying the grain growth for the pellets. And the carrier concentration decreased at 300 ℃ and increased again for high annealing treatment pellets.
Point 4: I guess author over/under estimated the ZT value because electrical conductivity and thermal conductivity were measured in two different direction. Author should correct this in the revised manuscript. What is the reproducibility of the data? I am not sure about the ZT value of this work in the present form because of the anisotropic structure of Bi2Te3.
Response4: Our text on this part was not clear. We removed the controversial issue and added into the experiment part.
Removed text: The electrical conductivity and Seebeck coefficient were measured via in-plane direction and thermal conductivity was measured by out-of-plane direction.
Added text in experiment part (line 70~72): Thermoelectric properties (electrical and Seebeck coefficient) of the pellets were measured in in-plane direction, which possess high performance in that direction [37].
The electrical conductivity and thermal conductivity have anisotropic effects and the Seebeck is almost isotropic effect. The anisotropic thermoelectric properties are relevant to the structural anisotropy of the samples. In my samples according to the grain realignment and grain growth, the texture changed from (0 0 l) and to random. It is regarded that the thermoelectric properties are not much affected by anisotropic characteristic (For 300 and 400 ℃ samples) due to the random alignment of texture. As the annealing temperature increased to 500 ℃, the grain grew and highly aligned to (0 1 5) plane. The electrical conductivity at (0 1 5) texture will be higher than the (0 0 l) aligned the structure. The texture change and intensity were displayed in Fig. S3. And thermal conductivity will be accordingly lower owing to the porous structure not by texture changed. In conclusion, the ZT value at 500 ℃ will be higher than other samples via supported electrical and thermal conductivity.
Point 5: Why author did not increase the temperature more than 500-degree C for their study?
Response 5: The melting point of bulk Bi2Te3 is around 580 ℃, which is lower than other alloy compounds. In addition, we assumed the micro-size of the Bi2Te3 powders may be lower than bulk materials. Thus, we applied the temperature not higher than 500 ℃.
We hope that you find the revised manuscript acceptable for the journal and look forward to hearing from you in due course. Thanks for your time and consideration.
Sincerely,
Yong-Ho Choa

Reviewer 2 Report
1. In Figure 5 not possible to see axis captions in line "C". It is need to increse size of text.
2.To correctly interpret the Seebeck effect, the electrical and thermal conductivity of the sample must be measured in the same direction. Otherwise, the interpretation of the results may be unreliable. It is clear that it is difficult to measure the thermal conductivity along the washer in the washer, but then the authors should give data on the electrical conductivity across the washer, in the direction in which the thermal conductivity was measured. Then we can talk about the isotropy of the results.
Author Response
Date: June 4, 2022
Dear Editor,
I am pleased to submit a revised manuscript for publication in Materials. The title of the manuscript and the names of the authors are as follow:
Microstructure evolution in plastic deformed bismuth telluride for the enhancement of thermoelectric properties
Haishan Shen , In Yea Kim , Jea Hong Lim, Hong-Baek Cho , Yong-Ho Choa
We found that comments of the reviewers were very helpful in reshaping the paper. We revised the manuscript accordingly. Below is the list of our responses to the comments.
Responses to Reviewer’s comments
Point 1: In Figure 5 not possible to see axis captions in line "C". It is need to increse size of text.
Response 1: Thnak you for your comment. According to reviewer’s comments, we increase text size.
Point 2: To correctly interpret the Seebeck effect, the electrical and thermal conductivity of the sample must be measured in the same direction. Otherwise, the interpretation of the results may be unreliable. It is clear that it is difficult to measure the thermal conductivity along the washer in the washer, but then the authors should give data on the electrical conductivity across the washer, in the direction in which the thermal conductivity was measured. Then we can talk about the isotropy of the results.
Response 2: Thank you for your comment.
Mostly the samples were cut like above figure to measure anisotropic effect. But my pellet with below 1 mm thickness was highly difficult to cut to cross plane samples.
Also we roughly measured the resistance with multi-meter from cross plane. The resistances are 2.0 Ω (RT sample), 0.3~0.4 Ω (300 ℃ sample) and 0.1~0.2 Ω (400 and 500℃ sample). It is not clearly exhibited the cross-plane electrical conductivity but roughly compared the electrical conductivity with cross-plane. The electrical conductivity in cross-plane will be highly increased by grain growth and combination between two particles via thermal treatment.
Overall, in my sample the texture changed from (0 0 l) to random and to (0 1 5) according to the grain realignment and grain growth. Due to anisotropic nature of Bi2Te3 materials, electrical conductivity on materials like aligned in (0 1 5) will be higher than the materials on highly oriented in (0 0 l). And thermal conductivity was lower by the porous structure not by texture change in my case. However, the Seebeck coefficient does not depend on the texture change and porous structure according to isotropic property. In conclusion, the ZT value on 500 ℃ sample (with main (0 1 5) plane texture and porous micro-structure) will be higher than other samples via supported electrical and thermal conductivities.
Sincerely,
Yong-Ho Choa

Reviewer 3 Report
Microstructure evolution in plastic deformed bismuth telluride for the enhancement of thermoelectric properties.
This work fabricates a highly dense compacted thermoelectric pellet using the waste scraps by cold pressing, and the thermoelectric properties are improved through high-temperature annealing. The ZT value is up to 1 at room temperature. I agree with the publication of this work after minor corrections.
1. After the thermoelectric pellet is annealed at high-temperatures (500 ℃), the grain size increases and internal pores are produced. This leads to a boost in ZT. While at low annealing temperatures, grain growth was not accompanied by an increase in pores. What is the physical mechanism behind this and why is it 500 ℃ and not some other temperature.
In addition, will higher temperature (over 500 ℃) annealing further improve ZT?
2. After high-temperature annealing, the density of the prepared thermoelectric sheet decreases, why does the electrical conductivity keep increasing?
3. In Fig. 5(c), the widths of the column are inconsistent across annealing temperatures. It is recommended that the four graphs be consistent for easy comparison.
4. The temperature should be given in Table 1.
5. I can’t find the supplementary figures in the supplementary file.
6. Some literatures need to be addressed: Physica Status Solidi B: Basic Solid State Physics, 1800442, (2019); Materials 2019, 12(20), 3453.
Author Response
Date: June 4, 2022
Dear Editor,
I am pleased to submit a revised manuscript for publication in Materials. The title of the manuscript and the names of the authors are as follow:
Microstructure evolution in plastic deformed bismuth telluride for the enhancement of thermoelectric properties
Haishan Shen , In Yea Kim , Jea Hong Lim, Hong-Baek Cho , Yong-Ho Choa
We found that comments of the reviewers were very helpful in reshaping the paper. We revised the manuscript accordingly. Below is the list of our responses to the comments.
Responses to Reviewer’s comments
Point 1: After the thermoelectric pellet is annealed at high-temperatures (500 ℃), the grain size increases and internal pores are produced. This leads to a boost in ZT. While at low annealing temperatures, grain growth was not accompanied by an increase in pores. What is the physical mechanism behind this and why is it 500 ℃ and not some other temperature. In addition, will higher temperature (over 500 ℃) annealing further improve ZT?
Response 1: Thank you for your comment.
Firstly, annealing process covers three main stages: (1) recovery stage, (2) recrystallization stage and (3) grain growth stage. At the low temperature such as 300 ℃, the materials may be in the recovery stage, which relieved internal stress and removed microcracks interface (See Fig. 3b). When the annealing temperature was above the recrystallization temperature (for example 400 ℃), the recrystallization, such as, texture realignment and change, was happened along with some grain growth. At high temperature (500 ℃), the material obtained enough energy and the atomic movement became too rapid to cover recovery, recrystallization and grain growth stage. The fully developed the new grains are mostly aligned to (0 1 5) plane, which has good electrical conductivity. In addition, rapid grain realignment and grain growth lead to porous structure, which lead to the lowest the thermal conductivity.
Secondly, the melting point of bulk Bi2Te3 is around 580 ℃, which is lower than other alloy compounds. Moreover, we assumed the micro-size of the Bi2Te3 powders may be lower than bulk materials like general properties of nano- to- micro-sized materials. Thus, we applied the temperature not higher than 500 ℃.
Point 2: After high-temperature annealing, the density of the prepared thermoelectric sheet decreases, why does the electrical conductivity keep increasing?
Response 2: Thank you for your comment. As the grain realignment and grain growth, the texture changed from (0 0 l) to random and then to (0 1 5) planes. Due to anisotropic nature of Bi2Te3, electrical conductivity on materials which is aligned into (0 1 5) direction will be higher than the materials highly oriented into (0 0 l) direction. And thermal conductivity was lower by the porous structure. However, the Seebeck coefficient is not dependent on change of texture and porous structure according to the isotropic property. In conclusion, the ZT value on 500 ℃ sample (with main (0 1 5) plane texture and porous micro-structure) will be higher than other samples via supported electrical and thermal conductivities.
Point 3: In Fig. 5(c), the widths of the column are inconsistent across annealing temperatures. It is recommended that the four graphs be consistent for easy comparison.
Response 2: Thank you for your comment. In order to comply with the referee’s comment, we modified the graphs to be consistent.
Point 4: The temperature should be given in Table 1.
Response 2: Thank you for your comment. In order to comply with the referee’s comment, we added the text about the properties which was obtained at room temperature in the table title.
From: Table 1. Comparison of thermoelectric parameters of n-type Bi2Te3 pellet with various powder processing methods.
To: Table 1. Comparison of thermoelectric parameters of n-type Bi2Te3 pellet with various powder processing methods. All the thermoelectric properties and ZT measured and obtained at room temperature.
Point 5: I can’t find the supplementary figures in the supplementary file.
Response 2: We apologize our mistake. We uploaded the supplementary file and added additional supportive data.
Point 6: Some literatures need to be addressed: Physica Status Solidi B: Basic Solid State Physics, 1800442, (2019); Materials 2019, 12(20), 3453.
Response 2: We added the previous references.
We hope that you find the revised manuscript acceptable for the journal and look forward to hearing from you in due course. Thanks for your time and consideration.
Sincerely,
Yong-Ho Choa

Round 2
Reviewer 1 Report
Authors have answered all of my questions and modified their manuscript accordingly. Thus, manuscript can be accepted in the present form.